# Trends in Neonatal Mortality at Princess Marie Louise Children’s Hospital, Accra, and the Newborn Strategic Plan: Implications for Reducing Mortality in Hospital and the Community

**DOI:** 10.3390/children10111755

**Published:** 2023-10-29

**Authors:** Edem M. A. Tette, Edmund T. Nartey, Mame Yaa Nyarko, Abena K. Aduful, Margaret L. Neizer

**Affiliations:** 1Department of Community Health, University of Ghana Medical School, Accra P.O. Box 4236, Ghana; 2Princess Marie Louise Children’s Hospital, Accra P.O. Box GP 122, Ghana; maame-yaa.nyarko@ghs.gov.gh (M.Y.N.); margaret.neizer@ghs.gov.gh (M.L.N.); 3Centre for Tropical Clinical Pharmacology and Therapeutics, University of Ghana Medical School, Accra P.O. Box 4236, Ghana; etnartey@ug.edu.gh; 4Family Medicine Department, Korle-Bu Teaching Hospital, Accra P.O. Box GP 4236, Ghana; adiepena@rt.gcps.edu.gh

**Keywords:** neonatal deaths, newborn strategic plan, neonatal sepsis, jaundice, hospital-based mortality

## Abstract

Background: In low and middle-income countries, close to half of the mortality in children under the age of five years occurs in neonates. Objectives: We examined the trend, medical conditions and factors associated with newborn deaths at the Princess Marie Louise Children’s Hospital (PML), Accra, from 2014 to 2017 (4 years). Methods: The study was a cross-sectional study. Data on age, sex, date of admission, date of discharge, cause of death and place of residence of these babies were obtained from the records department. This was transferred into an Access database and analyzed. Components of the Newborn Strategic Plan implemented at the hospital were described. Results: Neonatal sepsis, pneumonia and kernicterus were the major causes of death. Admissions increased and 5.4% of the neonates died, declining from 6.5% in 2014 to 4.2% in 2017 due to deliberate actions to reduce neonatal death. The highest mortality occurred in babies residing in an area more than 1 hour’s drive away from the hospital. Conclusion: Implementing the Newborn Strategic Plan was associated with a drop in mortality. A preponderance of community-acquired infections was observed. Thus, locality-specific interventions targeted at known determinants and implementing the newborn strategic plan are essential for reducing neonatal mortality.

## 1. Introduction

Although reducing neonatal deaths has been a matter of major public health concern for the past two decades, and neonatal death rates have slowly declined. Currently, neonatal deaths constitute almost half, 47%, of the deaths in children under the age of 5 years [1,2,3,4]. Back in 2015, newborn deaths were reported to constitute 45% of all deaths in children under the age of five years and this was projected to rise to 52% by 2030 [5]. In 2019, 2.4 million children died globally in their first month of life with children born in sub-Saharan Africa and Southern Asia carrying a 10 times higher risk of death than children born in high-income countries (HICs) [4]. This is because children in their first 28 days of life or the neonatal period, face the greatest risk of death from prematurity, birth asphyxia, infections, birth defects and kernicterus, among others, most of which are modifiable [4,6,7,8,9]. While interventions targeted at events in the intrapartum period are reported to result in 41% improvements in survival, care of small and sick neonates can also lead to a 35% improvement in survival [9]. Thus, the quality of health services available for the care of newborns is an important determinant of outcome [9,10,11,12].

In HICs where neonatal mortality is around 1%, this has occurred as a result of the development and provision of neonatal services including intensive care [1]. Apart from the lack of intensive care in many low- and middle-income countries (LMIC), neonatal deaths are also linked to poor access to health care, clean water, environmental sanitation and cultural practices relating to cord care which increase the risk of infection and jaundice [11,13,14,15]. Thus, efforts to prevent deaths in hospitals must occur concurrently with efforts to prevent neonatal diseases in the community [14,15,16]. The Sustainable Development Goal 3.2 (SDG 3.2) seeks to end preventable deaths of newborns and children under five years of age by 2030 [17]. As per this goal, countries must decrease their Neonatal Mortality Rates to 12 or lower deaths per 1000 live births. To achieve this reduction, the World Health Organization (WHO) and its Partners launched the Newborn Strategic Plan, initially spanning 2014–2018, globally [3,18,19]. Prior to this time, initiatives implemented to address under-five mortality in Ghana were based on Ghana’s Child Health Policy (2007–2015) which mainly focused on reducing mortality in older children [18]. Data on neonatal deaths in hospitals were limited and universal standards of care were largely unavailable [1,2,20].

The Newborn Strategic Plan 2014–2018 was launched in Ghana in July 2014, to provide the framework for a targeted strategy for accelerating the slow decline of neonatal deaths [18]. Its main objectives were to improve and increase training in essential newborn care among health professionals; provide basic neonatal resuscitation for adverse post-partum events; and improve the care of preterm/low birth weight babies and management of neonatal infections [18,19]. A review of the 14 implementation points of this strategy found deficiencies in the accuracy of the data available, making it difficult to assess progress [19]. The report also found that a number of deaths that had occurred in the community were unaccounted for. Facilities like the Princess Marie Louise Children’s Hospital which lack a delivery suite and primarily admit neonates from the community can provide useful information about these neonates. Neonatal sepsis is reported to be more common in out-born babies and community-based studies than in facility-based studies [15,21]. Current preventive measures include the use of Chlorhexidine for cord dressing, applying infection prevention and control measures, and preventing and treating maternal infections in pregnancy [22,23]. The latter can also influence stillbirths because, while in HICs, stillbirths are associated with conditions such as maternal obesity, inherited coagulopathies, diabetes and pregnancy-induced hypertension, infections dominate in LMICs occurring in about 50% of cases [23,24]. Implementing protocols for preventing, screening and treating ascending infections, syphilis, malaria and viral infections in pregnancy can reduce stillbirths and neonatal sepsis [22,23]. 

Guidelines for managing ascending infection due to Perinatal Group B *Streptococcus* Infection (GBS or *Streptococcus agalactiae*) are well established in HICs like USA [22,25]. Currently, the American Academy of Pediatrics (AAP) requires that maternal intrapartum prophylactic antibiotics such as penicillin, ampicillin, or cefazolin be given, based on screening results carried out at 36 0/7 to 37 6/7 weeks of gestation and newborns at risk should be evaluated for early-onset disease [25]. Similarly, the Centers for Disease Control (CDC) recommends that all women be given the Pertussis vaccine Tdap between the 27th to 36th week of gestation to reduce the risk of whooping cough [26]. This is because the majority of deaths occur in the age group 0–3 years, furthermore, outbreaks are frequent, and diagnosis can be challenging in neonates [26,27]. Unfortunately, these interventions are currently not widespread in LMICs like Ghana but need consideration. Since locality-specific interventions targeted at known determinants must go hand-in-hand with efforts to implement the newborn strategic plan to increase impact, this study examined the trend and medical conditions associated with neonatal mortality at PML from 2014 to 2017 and described the implementation of the NSP at the hospital.

## 2. Materials and Methods

### 2.1. Study Design

The study was a cross-sectional study, involving a records review of neonates who died at the hospital between 1 January 2014 and 31 December 2017, spanning a period of 4 years. Implementation of the Newborn Strategic Plan at the hospital was described and the mortality pattern by place of residence was displayed using GIS mapping.

### 2.2. Study Area

Princess Marie Louise Children’s Hospital is located at the commercial center of the capital, Accra, and provides primary and secondary care services to sick children either brought there by their parents (self-referred) from the community or referred from other health facilities. The hospital is the largest public children’s hospital in Ghana, and, historically, it has been credited with the diagnosis of Kwashiorkor. Currently, it has the largest nutritional rehabilitation center in the southern part of the country. Daily attendance at the Outpatient Department (OPD) and Emergency Room (ER) is about 152 patients, with an average of 2168 admissions six-monthly. It has no maternity services on-site, though it receives referrals from other facilities including James Town Maternity Home which is close by. At the beginning of the study, the hospital was a 74-bed hospital, but it currently has a bed capacity of 110. Since, originally, its primary focus was to provide nutritional rehabilitation services, there was no separate ward for neonates until 2009 when a special ward was created for babies under six months in the main block of the hospital. The babies’ ward was equipped with five baby cots, eight infant cots, three incubators and two locally made phototherapy units and only provided special care. Before then all babies who required admission were referred to other health facilities; thus, very sick newborns were generally not admitted to the hospital. 

### 2.3. Implementing the Newborn Strategic Plan at the Princess Marie Louise Hospital

In August 2018, PML Hospital created a special ward, known as the Newborn Care Unit (NBCU), for the care of newborns which became operational in 2019 to promote quality care and reduce mortality in the immediate post-natal period. The Unit is equipped with five incubators, five baby cots, one radiant warmer, one firefly and one lullaby phototherapy unit, one CPAP machine and three monitors (two fixed and one mobile unit). In addition, two other firefly phototherapy units were placed in the ER. The NBCU is not equipped to provide mechanical ventilation or parenteral nutrition. Services provided in this unit have strengthened the links between the hospital and other healthcare facilities that refer their babies to the unit. This has been achieved through Family Meetings with stakeholders, which is one of the key priorities of the Newborn Strategic Plan. 

Throughout the period of implementation of the NSP, the hospital has continued to create awareness about neonatal health conditions that increase the risk of neonatal deaths. These include prematurity, neonatal jaundice and sepsis. On World Prematurity Day on the 17th of November each year, staff embark on community sensitization on issues affecting premature infants, and advocate for good practices that improve their outcomes. The hospital also joined the Pediatric Society of Ghana to celebrate Neonatal Jaundice and Birth Asphyxia Awareness Months in May and September, respectively, each year. Through these advocacy drives, three firefly phototherapy units were donated to the facility. The month-long activities included community advocacy and social mobilization programs aimed at garnering support for the appropriate care of newborns affected by these conditions. 

Facility-based training was also undertaken to equip staff with the requisite knowledge and skills to manage newborns so affected including newborn resuscitation. Table 1 provides a summary of the key activities undertaken to improve neonatal care at PML since the Newborn Strategic Plan was launched. In 2015 the clinical team decided to prioritize the newborn at the Out-Patients Department (OPD) in response to mortality reports and other observations. All newborns were triaged on arrival, and, if very sick, were taken immediately to the ER for assessment and treatment. A decision was also made for all newborns to be seen by doctors’ preferably senior ones if available, rather than by a physician assistant. Training sessions were conducted for staff on common newborn conditions and how to manage them as well as Helping Babies Breath (HBB) an American program on neonatal resuscitation [28]. Facilities that refer to the hospital were invited to ‘family meetings’ during which feedback was given on referrals and recommendations on how to improve their pre-referral care were made, in order to reduce mortality. The acquisition of two firefly phototherapy units in the ER helped in managing neonatal jaundice. Triage was performed at regular intervals at the OPD.

Regarding breastfeeding, mothers were counseled on appropriate breastfeeding practices by nutritionists, dieticians and public health nurses. Among the topics discussed were the benefits of breastfeeding, proper attachment and positioning of the baby, how to express and store breast milk and problems of breastfeeding. This was conducted at the child welfare clinic (CWC), in the out-patients department, on the wards, in the dietician’s office, at the nutritional rehabilitation center and in the community using a chart by UNICEF. Training of dietician interns, students and new staff was also carried out. All staff received training on breastfeeding at educational meetings held at the hospital during the breastfeeding week. A nursery was created for breastfeeding and care of the children of staff during the period. However, a breastfeeding support group in the community was only established recently, in 2022. Mothers also received counseling on complementary feeding, family planning and other issues pertaining to child health. 

### 2.4. Study Population and Sampling

All newborns aged 0 to 28 completed days who were admitted to the Princess Marie Louise Children’s hospital and died during the study period were included in the study. They were excluded if their data were not captured by the records department or were incomplete. Consecutive patients who died at the hospital were recruited. 

### 2.5. Data Collection Instruments and Methods

The data were obtained from the computerized records of the hospital’s records department. Data on age, sex, date of admission, date of discharge, causes of deaths, place of residence and total admissions were obtained. Data on birth weight and gestational age was not collected routinely at the time and therefore were not among the records.

### 2.6. Data Analysis

The data were entered into a computerized record form using a Microsoft Access (Microsoft Corporation, One Microsoft Way, Redmond, WA 98052-6399, USA) program and cleaned. Analysis was performed using Stata SE 14.0^®^ (Stata Corporation, College Station, TX, USA). Conditions associated with mortality were summarized using descriptive statistics such as frequencies and proportions. Means with standard deviation were also computed. Neonatal deaths were reported as a proportions of neonatal admissions. The place of residence was plotted using a GIS database and summarized in graphs. The mean distance and standard deviation were also determined.

### 2.7. Ethical Approval

Ethical Approval for the study was obtained from the Ghana Health Service Ethical Review Committee from 2014 to 2017 (GHS-ERC 09/03/2017) which waived the need for patient’s consent. Permission was sought from Princess Marie Louise Children’s Hospital and granted. Confidentiality was ensured.

## 3. Results

Out of a total of 21,185 admissions at the hospital, 1988 of the admissions were neonatal admissions and 108 neonates died; thus, the proportion of neonatal deaths per total neonatal admissions was 5.4%. Neonatal admissions rose from 446 neonatal admissions in 2014 to 508 admissions in 2015 and 607 admissions 2016, but it declined to 427 neonatal admissions in 2017. Age at admission for the neonates who died at PML hospital ranged from 1 day to 21 days with a mean age of 6 days (SD = 5.3). The majority of neonates who died were males (57%, n = 62), while 43% (n = 46) were female. The majority of the children who died (59%, n = 64) did not have health insurance. 

Table 2 summarizes the cause of death of newborns attending PML from 2014 to 2017. The top five causes of death of patients admitted were suspected sepsis, pneumonia, kernicterus, birth asphyxia and anemia, comprising 3.6%, 0.7%, 0.6%, 0.4% and 0.3% of admissions, respectively, and contributing 65.7%, 12.0%, 11.1%, 7.4% and 4.6%, respectively, of the 108 neonatal deaths. Thus, altogether, 90 (83.3%) had at least one of the first five causes of death. Congenital malformations and meconium aspiration were among the next most common causes of death. Three patients died from a bleeding disorder. Low birth weight and prematurity were not among the major causes of death. 

The figure below (Figure 1) displays the trend of the proportion of neonatal deaths per each year. Altogether, the proportion of neonatal deaths among the babies admitted was 6.5% (29/446) in 2014. It rose to 7.7% (39/508) in 2015, declined sharply to 3.6% (22/607) in 2016 and rose again to 4.2% (18/427) in 2017.

The figure below (Figure 2) displays the trend of the five common causes of neonatal deaths in the hospital over the four-year period from 2014 to 2017. Neonatal sepsis showed a mild decline from 2014 to 2015, then it decreased sharply from 2015 to 2016 and tapered down over the next year. Pneumonia and kernicterus showed fluctuating trends over the four-year period, while birth asphyxia and anemia showed a stable pattern with a slight rise in anemia from 2016 to 2017. 

Many of the 71 babies with a diagnosis of neonatal sepsis also had other diagnoses, but the most frequent was neonatal jaundice occurring in 27 babies, including 8 of the babies with kernicterus, 9 babies with pneumonia, 5 babies with birth asphyxia and 4 babies with anemia, among others. Neonatal jaundice was a prominent finding during the study. In all, 39 babies were reported as having neonatal jaundice, including all 12 babies with kernicterus, 20 babies with other diagnoses specified as causes of death in Table 2 and 7 babies whose causes of death were labelled as unspecified as they did not have any other diagnosis. 

Table 3 below shows the time spent in the hospital before demise. It ranged from 1 day to 18 days with a mean of 2 days (SD = 2.3).

The mean distance from the places of residence to the hospital of the neonates who died was 13 km (SD = 10.6), and it ranged from 0.2 km to 29 km (Figure 3). Patients from Kasoa in the central region had the highest mortality related to their place of residence. There seemed to be some clustering of deaths within 4 km radius of the hospital. However, beyond the 4 km radius, it appeared the deaths were similarly distributed. 

## 4. Discussion

There was a rise in admissions during the study period except the period from 2016 to 2017. This rise has been observed in other studies [12,29]. Rising admissions can increase neonatal deaths, if steps are not taken to address the increasing workload, logistics and human resource requirements that ensue, as well as providing sufficient space to prevent overcrowding and the spread of infections [12,30]. Although the rise in admissions started before the introduction of the Newborn Strategic Plan (NSP) [20], some of this rise may have accompanied the increase in public awareness from educational activities undertaken as part of the NSP or the Paediatric Society’s initiative to curb neonatal jaundice [31]. While this may have saved many babies, managing conditions like neonatal jaundice can be resource-intensive, especially in severe cases that require exchange transfusions and special phototherapy. Thus, when carrying out awareness-creation programs and encouraging early presentation, it must be accompanied by more equipment and training of personnel to match the increasing workload from the additional admissions that will ensue. That is why it is gratifying to note that a new newborn unit was commissioned at PML with special equipment to treat neonatal jaundice as part of the NSP. 

Data on neonatal admissions were available in 2014–2017 due to the improved collection of data as part of the Newborn Strategic Plan [19]. The overall proportion of neonates who died in hospital was 5.4% which is lower than the proportion who died in a study at the Upper West Regional Hospital (UWRH) in Ghana (8.9%), Tamale Teaching Hospital in Ghana (13.4%), Kenya (10.2%), Nigeria (10.1%), Cameroon (15.7%), and Malawi (20%) [6,32,33,34,35,36]. This is possibly because while large hospitals and teaching hospitals are better resourced, they also tend to admit more high-risk babies. In contrast, a study at a private tertiary hospital in Uganda reported that neonatal mortality was 5.7% after declining from 8.2% following selected interventions [12]. Therefore, benchmarks for assessing outcomes in these settings must consider facility type as well as context. 

The proportion of neonatal deaths fell through a fluctuating trend from 6.5% in 2014 to 4.2% in 2017. In contrast, the trend reported in a study conducted in Cameroon over a 7-year period from 2004 to 2010 followed a steady downward trend dropping from 12.4% to 7.2% [37]. Similarly, an overall downward trend was reported during the 5-yr period of study from 2013 to 2017 at the Tamale Teaching Hospital in Ghana [32]. The sharp fall from 2015 to 2016 coincided with the second year of the launch of the Newborn Strategic Plan and resulted in a reduction in overall deaths as well as deaths from sepsis. This is likely to be due to the deliberate efforts made to reduce newborn deaths through training, and changes in manpower that ensured that neonates were seen by more experienced staff and triaged. Although the Emergency Triage Assessment and Treatment (ETAT) protocol was applied on arrival, it was supplemented with intermittent triage of babies waiting in queues at the Out-Patient Department [38]. This ensured that babies whose condition changed were picked up and seen early. 

In most low- and middle-income countries, prematurity, intrapartum-related deaths such as birth asphyxia, and infections are the three major causes of neonatal deaths [9]. In this study, neonatal sepsis, kernicterus and pneumonia were the three most common causes of mortality. This suggests that community-acquired infections underlie mortality in this setting. Furthermore, unlike studies conducted in facilities with delivery suites, prematurity was not among the top five diseases in this study and birth asphyxia was also not among the top three diseases [32,33,34]. The diagnosis of sepsis was mainly clinical and was underlying 65.7% of the neonatal death. This proportion was much higher than the report from UWRH (26.3%) [6], Nigeria (17.8%) [34] and Malawi (23%) [36] possibly because our patients were mostly out-born babies and tended to present with more infections. Although this was a facility-based study, our findings are similar to the results of a systematic review and meta-analysis on the global incidence and mortality from neonatal sepsis which found that the incidence of neonatal sepsis was >4 times higher in community-based studies compared with facility-based studies [21]. Identification and treatment of newborns with infections is reported to be suboptimal in many LMICs and compounded by traditional practices, financial challenges, poor health systems and antimicrobial resistance [10]. Altogether, 59% of the children did not have health insurance, which could have been a barrier to treatment. 

Finding that neonatal jaundice was common corroborates with a study carried out in Cape Coast Teaching Hospital in Ghana [39], and its association with neonatal sepsis is well-recognized [15]. Though not typically a cause of death, severe neonatal jaundice can cause kernicterus, which is a risk factor for death and neurologic damage resulting in cerebral palsy [40,41]. ABO and Rhesus blood group incompatibility, G6PD deficiency, sepsis and low birth weight are the common causes of neonatal jaundice [15,42,43]. Preventing deaths must include education on the danger signs of neonatal jaundice and sepsis, improved care, health insurance registration and early presentation, as well as identifying practices within the community that aggravate these diseases and stopping them.

The male sex has been found to be a predictor of neonatal death in a study by Hoque et al. in South Africa [44], who found that 63% of the neonatal deaths there were males, similar to the 57% recorded in this study. The majority of neonatal deaths occur in the first week of life and we also found that 96.3% of neonatal deaths occurred in the first week of life which was slightly higher than findings in Tanzania (87.5%) [45] and Cameroon (74.2%) [37]. As occurs in most countries and in other parts of Ghana [39], the majority of the deaths, 42.6%, in this study occurred in the first 24 hours of life. This is usually related to the severity of the condition, the mode of transport, the condition of the neonate on arrival and the quality of care received by the mother and baby during labor and at birth [9,10,46].

The GIS map shows that the hospital serves a wide catchment area. Apart from some clustering of deaths within a 4 km radius of the hospital which might have been from the large maternity home in the area, the distribution of mortalities appeared similar across the areas within the other perimeters. Thus, preventive activities should extend beyond communities in the immediate catchment area of the hospital. It was disappointing to note that a high proportion of the neonates who died came from Kasoa in the Central Region, located 23.5 km (1–2 hours’ drive) from the hospital. This is because, in 2013, it was identified as the place of residence associated with the highest number of deaths in children under the age of five years [47]. Studies have found that poor geographic accessibility of health facilities increases the risk of newborn deaths [48,49]. As previously recommended, providing access to newborn units closer to the residences of these patients and safer transfer processes for those needing referral are likely to reduce neonatal deaths in this setting [50]. Nevertheless, it is unclear why larger facilities were bypassed to transfer some of these neonates to PML, suggesting the referral system needs to be evaluated. A study in the Northern Region of Ghana reported a reduction in neonatal mortality following better infection control and treatment practices as well as the creation of NICUs and more Community-based Health Planning and services (CHPS) zones for treating sick neonates [16]. This stresses the importance of prioritizing different interventions to suit the local context.

Although the Newborn Strategic Plan has brought about a reduction in the proportion of neonatal deaths at PML, additional interventions are needed to reduce mortality further. Interventions such as total parenteral nutrition, mechanical ventilation and using body cooling devices can reduce mortality further, but are part of intensive care and are currently recommended for tertiary care facilities in Ghana such as regional and teaching hospitals. Since PML is considered a Level II (District Level) facility, current guidelines stipulate that it operates a Special Care Newborn Unit (SCNU) as it is doing now [51]. Thus, mechanical ventilation is accessible at Greater Accra Regional Hospital, a tertiary facility located 5.4 km from PML hospital with the capacity to ventilate eight babies. Admission is, however, dependent on the availability of space, though caregivers pay for the service. Parenteral nutrition, on the other hand, is not available. Consequently, to achieve Universal Health Coverage for all (SDG Target 3.8) and for access to be equitable, the government has to provide more resources and free medical care for all newborns irrespective of their mother’s NHIS status. Also, periodic reviews of newborn outcomes will inspire continuous improvements to lower mortality rates and, perhaps, justify making a case for upgrading the service to intensive care.

Apart from providing special newborn care services, the hospital also offered general pediatric out-patient clinics, emergency care for medical conditions, pediatric surgery and dental care during the period under review. It held specialist clinics for children with asthma, sickle cell disease, HIV/TB and neurodevelopmental conditions; other services were dietetics, nutritional rehabilitation, physiotherapy and nurse-led Ear, Nose and Throat (ENT) clinics, Eye clinics and Child Welfare Clinics. The CWC offered immunization, growth monitoring, vitamin A and counseling on infant feeding, among others. Maternal services such as family planning and clinical care for mothers with HIV were also delivered in accordance with Ghana’s Child Health Policy and other maternal and child health policies [51,52,53,54].

Some limitations were encountered during the study. We were unable to compare neonates who survived with neonates who died and classify the data in terms of birth weight and gestational age. Furthermore, the diagnosis of sepsis was mainly clinical as blood cultures were performed only on patients who could afford it.

## 5. Conclusions

The study found that despite rising neonatal admissions during the 2014–2016 period, overall mortality was relatively low compared to other similar settings, and a sharp decline in mortality occurred due to deliberate actions to reduce neonatal deaths. Sepsis, pneumonia and kernicterus were the most common causes of death, highlighting the importance of community-acquired infections in the etiology of neonatal deaths in this setting and the need for interventions to reduce them. Neonatal jaundice was prominent. Overall, implementing the Newborn Strategic Plan led to better data capture, training of staff, more resources and programs to raise awareness at the hospital and in the community, but more importantly, it stimulated conscious efforts to prevent neonatal deaths within the hospital. Nevertheless, further resources are needed for better diagnosis and treatment of infection; continued public education on hygiene and danger signs of neonatal conditions; and upgrading the care and providing universal health care for these babies. Improving water sources and environmental sanitation in communities, creating more neonatal units, especially one close to Kasoa, and improving access are needed, together with further studies to identify the causes and determinants of sepsis.

## Figures and Tables

**Figure 1 children-10-01755-f001:**
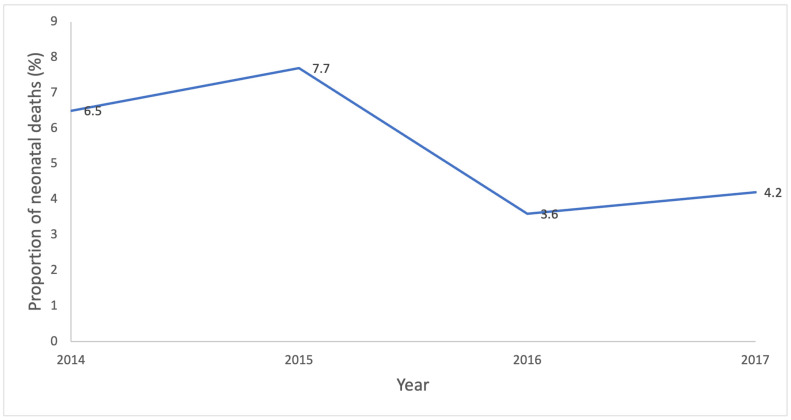
Chart showing trend of the proportion of neonatal deaths in each year.

**Figure 2 children-10-01755-f002:**
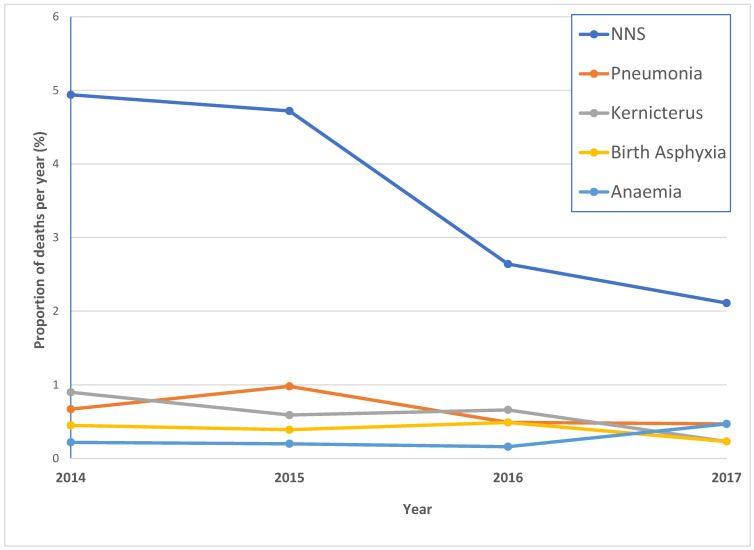
Mortality trend of the top five causes of neonatal mortalities (2014–2017). NNS = neonatal sepsis, NNJ = neonatal jaundice.

**Figure 3 children-10-01755-f003:**
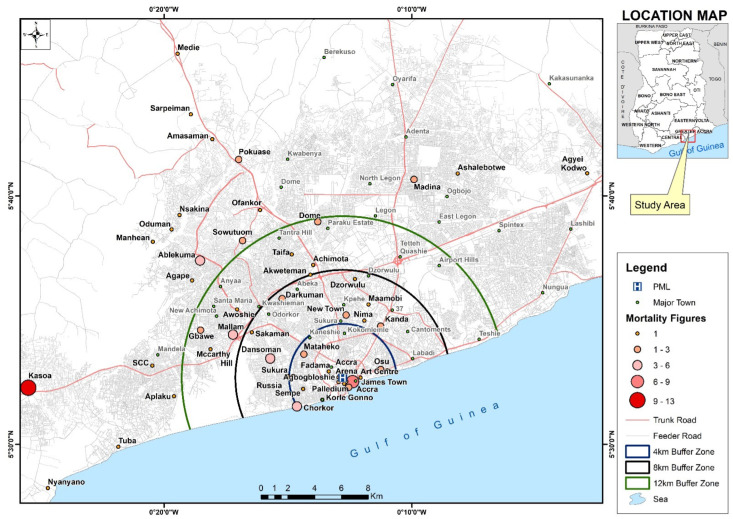
Distance from PML Hospital and residence of neonates who died in 2014–2017 (N = 103).

**Table 1 children-10-01755-t001:** Newborn strategic plan and measures taken to implement the 14 strategies from 2014 to 2018.

STRATEGY	KEY CHANGES 2014–2018
STRATEGY 1: Developing or updating necessary policies, standards and coordinating mechanisms to support newborn care activities	Aside from the initial assessment on arrival using the Emergency Triage Assessment and Treatment (**ETAT**) protocol, babies in the queue at the outpatient department were reassessed intermittently to see if their condition was still stable or had changed. Depending on the findings, appropriate measures were taken to stabilize them.
STRATEGY 2: Updating the National Health Information Management System/District Health Information Management System (DHIMS2) to include newborn indicators	Data on newborns were captured separately from other children. Segregating the data and using the newborn indicators highlighted the newborn health issues.
STRATEGY 3: Increasing health financing for newborn care	Newborns whose mothers were registered on the National Health Insurance Scheme (NHIS) were able to access care using their mother’s health insurance.
STRATEGY 4: Ensuring procurement, equitable distribution and maintenance of quality essential medicines, medical devices, and commodities for newborn care	A newborn Care Unit was set up in 2018 and furnished subsequently with 5 incubators, 5 cots, 1 radiant warmer, 1 firefly and 1 lullaby phototherapy unit, 1 CPAP machine and 3 monitors. Two (2) firefly phototherapy units were also placed at the ER.
STRATEGY 5: Ensuring availability and suitable distribution of key competent health workers	Nurses and doctors were sent to the Regional Hospital (GARH) to work in the Neonatal Intensive Care Unit (NICU) for a month to refresh their knowledge and skills in newborn care. After establishing the newborn unit, and following the acquisition of the CPAP machines, nurses were sent to learn how to care for the newborn on CPAP. Also, a pediatrician for GARH came to PML to train staff on the use of the machine
STRATEGY 6: Improving the capacity of facility-level health workers to address newborn care	Facility-based training was carried out to Help Babies Breathe, identify sick newborns and manage common health conditions such as neonatal jaundice, sepsis and birth asphyxia. Protocols for managing common newborn conditions were developed.
STRATEGY 7: Building the capacity of Community Health Workers to promote newborn health	No new activity conducted
STRATEGY 8: Promoting and institutionalizing quality improvement, including supportive supervision and mentoring	This was done by the district and regional team but often did not include newborn issues. Peer review is done annually to assess newborn issues like availability of equipment and drills corner for a “golden minute”, among others.
STRATEGY 9: Scaling up a strengthened and expanded Mother/Baby-Friendly Initiative	There was regular training of mothers at the hospital on breastfeeding. The national breastfeeding week was observed annually with advocacy and talks for mothers and staff.
STRATEGY 10: Strengthening advocacy, communication, social mobilization and other community-based interventions	Celebration of Neonatal Jaundice Day, World Prematurity Day, and Birth Asphyxia Awareness Month in conjunction with the Pediatric Society of Ghana.
STRATEGY 11: Strengthening links between health facilities and communities	Community advocacy and media engagements on neonatal issues were carried out as well as the ‘family meetings’.
STRATEGY 12: Strengthening public-private partnership	Donations of consumables and equipment were received from private organizations.
STRATEGY 13: Operationalizing an effective plan for monitoring and evaluation	The Newborn Strategy and Action Plan was disseminated in a clinical meeting.
STRATEGY 14: Managing the Newborn Strategy and Action Plan	A neonatal audit committee was set up to audit all neonatal deaths within 7 days of occurrence and make recommendations to prevent future occurrences. Mortality meeting were carried out regularly however, the reporting system needed further development.

**Table 2 children-10-01755-t002:** Cause of death of newborns attending PML from 2014 to 2017 in the 108 babies who died *.

	Total	2014	2015	2016	2017
Cause of death	N = 1988	N = 446	N = 508	N = 607	N = 427
	n,p	n,p	n,p	n,p	n,p
Neonatal sepsis	71 (3.6)	22 (4.9)	24 (4.7)	16 (2.6)	9 (2.1)
Pneumonia	13 (0.7)	3 (0.7)	5 (1.0)	3 (0.5)	2 (0.5)
Kernicterus	12 (0.6)	4 (0.9)	3 (0.6)	4 (0.7)	1 (0.2)
Birth Asphyxia	8 (0.4)	2 (0.4)	2 (0.4)	3 (0.5)	1 (0.2)
Anaemia	5 (0.3)	1 (0.2)	1 (0.2)	1 (0.2)	2 (0.5)
Aspiration	4 (0.2)	-	3 (0.6)	-	1 (0.2)
Congenital malformation	4 (0.2)	2 (0.4)	1 (0.2)	1 (0.2)	-
Meconium Aspiration	4 (0.2)	-	2 (0.4)	2 (0.3)	-
Bleeding Disorder	3 (0.2)	1 (0.2)	1 (0.2)	-	1 (0.2)
Gastroenteritis	3 (0.2)	1 (0.2)	1 (0.2)	1 (0.2)	-
Respiratory distress	3 (0.2)	-	1 (0.2)	2 (0.3)	-
Dehydration	2 (0.1)	1 (0.2)	1 (0.2)	-	-
Hypoglycaemia	2 (0.1)	-	1 (0.2)	1 (0.2)	-
Impetigo	2 (0.1)	-	-	2 (0.3)	-
Malnutrition	2 (0.1)	1 (0.2)	-	1 (0.2)	-
Meningitis	2 (0.1)	-	1 (0.2)	1 (0.2)	-
Prematurity	2 (0.1)	1 (0.2)	1 (0.2)	-	-
HIV (Positive/Exposed)	2 (0.1)	1 (0.2)	1 (0.2)	-	-
Drug withdrawal	1 (0.05)	-	1 (0.2)	-	-
Encephalopathy	1 (0.05)	-	-	1 (0.2)	-
Heart Disease	1 (0.05)	1 (0.2)	-	-	-
Hypothermia	1 (0.05)	-	-	1 (0.2)	-
Low Birth Weight	1 (0.05)	1 (0.2)	-	-	-
Unspecified diagnosis	7 (0.4)	1 (0.2)	3 (0.6)	2 (0.3)	1 (0.2)

P: Proportion of neonatal deaths in percentage (mortality rate) * multiple causes of death apply to some babies.

**Table 3 children-10-01755-t003:** Duration of admission before demise 2014–2017.

Duration of Admission (Days)	N	Percentage (%)
1	46	42.6%
2	38	35.2%
3	9	8.3%
4	5	4.6%
5	1	0.9%
6	4	3.7%
7	1	0.9%
>7	4	3.7%
Total	108	100.0%

## Data Availability

The data are available on request. The data used for this study are owned by and under the primary jurisdiction of the Ghana Health Service Ghana. Enquiries about using the data can be made to the Director General of the Ghana Health service.

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
