# Peer review of "Trends in Neonatal Mortality at Princess Marie Louise Children’s Hospital, Accra, and the Newborn Strategic Plan: Implications for Reducing Mortality in Hospital and the Community"

_children, 2023, doi:10.3390/children10111755_

Round 1

Reviewer 1 Report

I read your paper and I find the statistics quite disturbing when it comes to mortality rates in newborns.

It seems that your strategic plan worked out and brought some improvement but still, the mortality is high. You should aim at lowering the neonatal mortality rate below 3%.

I suggest adding some equipment for the Newborn Care Unit (NBCU): parenteral nutrition, mechanical ventilation, and body cooling devices because most newborns are born outside the hospital, and birth asphyxia is frequent.

I also suggest adding free medical care for all newborns no matter if their mothers are contributors to the National Health Insurance Scheme (NHIS)

Neonatal jaundice ( not kernicterus) should not appear as a main cause of death in 2023 no matter the country we are talking about. I understand death from kernicterus which is rare and still present in more developed countries, but jaundice should disappear as a cause of death.

The paper should emphasize the need for continuous death rate improvement in the neonatal population. Hope to see better numbers in the future!

Wish you all the best!

English is fine

Reviewer 2 Report

Dear editor,

Thank you for providing the opportunity to review the article titled "Trends in neonatal mortality at Princess Marie Louise Children's Hospital, Accra and the Newborn Strategic Plan: implications for reducing mortality in hospital and the community". The authors investigated the trend, medical conditions and associated factors causing newborn deaths at a local hospital. In this article, the development of the physical infrastructure and device park of the local hospital, where the study were conducted, was mentioned. The main factor that ensures the improvement of health indicators is related to increasing the access and quality of health services to the society with trained manpower rather than the development of physical infrastructure and equipment park. In the article, it should be discussed in detail whether neonatal resuscitation program (NRP) training is given to healthcare professionals dealing with infant health, and whether services related to maternal and infant health are developed (such as supporting breastfeeding). Since this study reflects the social structure of the region and the state of hospital services, I believe that it would be useful to publish it in the journals of that region in terms of being an example for the hospitals in the region.

Sincerely

Reviewer 3 Report

This a very interesting study describing the effects of a newborn mortality reduction plan implanted in Ghana. The study is well designed, the methods and the plan are well described and the results are adequately stated. It is believed that this manuscript contains relevant data that might be useful for other low and middle income countries to use as local policies to prevent newborn deaths.

Reviewer 4 Report

Dear authors

it has been a pleasure to go trough your paper

i think that every effort in reducing neonatal death should be supported

in low income countries difficulties sepsis appearead to be with jaunice and pneumonia the main reasons for NND

introduction may be improved

with

1) a mention regarding the fact that sepsis is also a main finding in also stillbirth therefore to carefully look and treat infections during pregnancy and childbirth is crucial (read and cite        doi: 10.36129/jog.2022.20)

2) please also mention guidelines regarding term prevention of streptococco beta agalactiae ascending infection prevention

3) please mention also guidelines regarding prevention of pertussis 

4) add a table with a summary of findings

best regards

please add a revision of English from a native english

Round 2

Reviewer 2 Report

Dear Authors,

I read the revised version of the article titled "Trends in neonatal mortality at Princess Marie Louise Children's Hospital, Accra and the Newborn Strategic Plan: implications for reducing mortality in hospital and the community".

I think that publishing the positive effect of improving infant health services on neonatal mortality data in the hospital where the research was conducted would be beneficial for hospitals in the region.

Sincerely.